# Distinct Phenotypic Variation of *Blastocystis* sp. ST3 from Urban and Orang Asli Population—An Influential Consideration during Sample Collection in Surveys

**DOI:** 10.3390/biology11081211

**Published:** 2022-08-12

**Authors:** Arutchelvan Rajamanikam, Ho Shiaw Hooi, Madhav Kudva, Chandramathi Samudi, Suresh Kumar Govind

**Affiliations:** 1Department of Parasitology, Faculty of Medicine, University of Malaya, Kuala Lumpur 50603, Malaysia; 2Department of Medicine, Faculty of Medicine, University of Malaya, Kuala Lumpur 50603, Malaysia; 3Gastroenterology and Hepatology Specialist Clinic, Pantai Hospital, Kuala Lumpur 59100, Malaysia; 4Department of Medical Microbiology, Faculty of Medicine, University of Malaya, Kuala Lumpur 50603, Malaysia

**Keywords:** *Blastocystis* sp., urban, orang asli, phenotype, pathogenicity

## Abstract

**Simple Summary:**

*Blastocystis* sp. is a common intestinal protozoan of humans with the phenotypic characteristics strongly associated with its activity, including pathogenicity. This characteristic varies, but the variation has not been clearly understood. The present study evaluates the variation when a single subtype of *Blastocystis* sp. was isolated from a population with distinct gut microbial composition, namely, the urban and orang asli(indigenous) population. *Blastocystis* sp. cells isolated from orang asli individuals had a higher growth rate with elevated resistance to harsh conditions. Distinct surface coats with amoebic forms were noticed in parasite cells from urban individuals. Proteases, commonly a virulent factor in other parasites, showed variation depending on the isolation source. Stimulation of cancer cell proliferation by only *Blastocystis* sp. isolated from urban individuals is suggestive of the variation at the antigenic level. This phenotypic variation suggests that implicating subtype to pathogenicity may be too early, and a deeper understanding of *Blastocystis* sp. and microenvironment interaction is essential.

**Abstract:**

*Blastocystis* sp. is a globally distributed protozoan parasite with uncertain pathogenicity. Phenotypic variation in *Blastocystis* sp. suggests its adaptation; however, the phenotypic features of *Blastocystis* sp. ST3 from a distinct source of isolation is unknown. *Blastocystis* sp. isolated from individuals in urban and orang asli (indigenous population in Selangor, Malaysia) settlements were studied for phenotypic characteristics such as growth profile, morphology, ultrastructure, and resistance to harsh conditions. Subsequently, pathogenic potentials, such as in protease activity and the ability to stimulate the proliferation of cancer cells, were assessed. Higher parasite counts with granular and apoptotic forms were found in *Blastocystis* sp. from orang asli individuals. Cells with fuzzy coats and amoebic structures which seemingly implicate increased interaction with bacteria were seen predominantly in urban symptomatic persons. Also, *Blastocystis* sp. from orang asli isolates resisted harsh environments, suggesting longer co-adaptation to the hosts. Urban and orang asli symptomatic isolates possessed a predominance of only cysteine protease, whereas all the asymptomatic isolates showed significantly higher cysteine, serine, or aspartic protease activity. However, only solubilized antigen from urban symptomatic isolates showed significant stimulation of cancer cell proliferation. For the first time, our findings demonstrate significant phenotypic variation in a single subtype, ST3 of *Blastocystis* sp., isolated from urban and orang asli populations that are known to have distinct gut microbial compositions. The outcome emphasizes the importance of identifying people’s locations and lifestyles during sample collection before forming conclusions on the prevailing data and implicating subtypes to pathogenicity. The environment plays a significant role in *Blastocystis* sp. infection.

## 1. Introduction

*Blastocystis* sp. is an intestinal protozoan parasite often linked to gastrointestinal symptoms such as diarrhea, bloating and flatulence, abdominal cramps [1], and Irritable Bowel Syndrome (IBS) [2,3], as well as extra-gastrointestinal illnesses such as urticaria [4,5] and iron-deficiency anemia [6,7]. However, its pathogenicity is still a question. A high prevalence of up to 100% has been identified in developing countries, compared to as low as 4% in developed countries [8,9]. High prevalence is often associated with compromised hygiene and sanitation levels.

Recently, a large-scale stool survey was conducted to compare intestinal parasitic infection in two distinct populations from Malaysia. It was found that the rural population had a greater prevalence of *Blastocystis* sp. Than the urban [10]. Other studies have reported a similar prevalence in the rural population ranging from 30% to 100% [9,11,12]. Several studies comparing the urban and rural microbiome have shown that greater bacterial diversity and distinct microbial metabolic activity was found typically in rural population [13,14]. Similarly, a study from Malaysia has demonstrated that the aborigines have a distinct gut bacterial composition [15]. The variations exhibited by *Blastocystis* sp., when isolated from these populations, are unexplored, and its adaptations are still unclear.

Conclusive evidence on the pathogenicity is lacking due to its extensive genetic diversity (emergence of varying subtypes) [16], cryptic host specificity [17], and its presence in healthy individuals [18]. Proteases reportedly contribute to the pathogenesis of *Blastocystis* sp., as in other gut protozoan parasites like *Entamoeba histolytica* [19,20]. Cysteine protease of *Blastocystis* sp. degrades IgA and elicits IL-8 inflammatory response [21,22,23]. Previously, elevation of protease activity has been demonstrated in *Blastocystis* sp. ST3 and in symptomatic isolates [24]. Some studies have also noted a higher percentage of proliferation in colon cell lines induced by antigens obtained from symptomatic *Blastocystis* sp. ST3 isolates. This suggests an alteration in antigenic content of symptomatic isolates [25,26]. However, the variation of proteases in *Blastocystis* sp. and the cancer cell proliferation due to *Blastocystis* sp. antigens when isolated from different populations have not been documented, although this knowledge will be an essential consideration in future epidemiological studies.

To date, findings on the pathogenicity of *Blastocystis* sp. have been contradicting and recent studies are inclined towards subtype-dependent pathogenicity [27]. However, the variation due to the source of isolation and within a single subtype has not been elucidated yet. The present investigation attempts to use *Blastocystis* sp. colonizing individuals from two distinct populations known to have different gut microbiota, namely urban and orang asli. We assessed if there exists phenotypic variation and difference in pathogenic potentials between the isolates despite retaining the same subtype. The variation seen is linked to corresponding adaptations by the organism.

## 2. Material and Methods

### 2.1. Blastocystis sp. Isolation and Genotyping

*Blastocystis* sp. were isolated from symptomatic and asymptomatic individuals from urban and orang asli population. Urban isolates were collected from patients visiting the Gastroenterology Unit, University Malaya Medical Centre and Gastroenterology and Hepatology Specialist Clinic, Pantai Hospital, Kuala Lumpur, Malaysia for non-specific gastrointestinal symptoms. These patients were regarded as symptomatic by the clinician. *Blastocystis* sp. isolates obtained from random fecal screening among healthy individuals around Petaling Jaya, Malaysia area were regarded as urban asymptomatic. These individuals were verified further upon questioning that they were all city dwellers.

For *Blastocystis* sp. isolates from orang asli, random fecal sample collection was carried out in three orang asli settlements on the outskirts of Selangor, Malaysia. Individuals in the orang asli population rarely visited medical personnel when GI symptoms are experienced. Hence, a questionnaire was used to assess the type of symptoms and their frequency to identify symptomatic individuals. Individuals experiencing non-specific GI symptom more than three times a week are considered symptomatic. The age group criteria for sampling were standardized to 20–40 years old without any organic bowel diseases such as cancer or inflammatory bowel disease (IBD).

Fecal samples were collected in a UV-sterilized stool container and screened within 6 h of collection. Fecal samples were screened for other gastrointestinal parasites using the formal ether concentration (FEC) technique [28]. About 50 mg of fecal sample was inoculated into 3 mL Jones’ medium supplemented with 10% horse serum as reported previously [29]. The parasite cultures were incubated at 37 °C and screened daily for 5 to 7 days. The presence of vacuolar forms of *Blastocystis* sp. was regarded as positive. Isolate with only *Blastocystis* sp. as the sole infectious organism was selected in this study. The parasite cultures were continuously maintained in in vitro and passaged once every 3 to 4 days. Basic aseptic techniques were employed throughout the maintenance of these parasite cultures.

DNA was extracted using the Macherey Nagel Soil DNA extraction kit following the manufacturer’s protocol without any modification. The DNA was extracted from the in vitro culture. The extracted DNA was used as a template to amplify the 18S small subunit ribosomal RNA gene (18S SSU-rDNA) at the length of 600 bp using the protocols and primers described previously [30,31]. Amplified products were sequenced and compared using the existing database in public databases for molecular typing and microbial genome diversity (PubMLST). The ST3 rDNA sequences were compared and phylogenetic analysis was carried out using CLUSTALW. For all the phenotypic analyses, five isolates of *Blastocystis* sp. ST3 from each group (urban symptomatic, urban asymptomatic, orang asli symptomatic, and orang asli asymptomatic) were randomly selected.

### 2.2. Growth Characterization

Day 3 parasites were counted using a hemocytometer chamber and inoculated into a 1 ml medium with the final concentration of 1 × 10^5^ cells/mL. The parasites were counted every day for ten days using the trypan blue exclusion test to determine the viability of cells. The peak cell growth, usually on day two or three, was compared statistically. The number of granular and amoebic forms per ml was counted. Generation time was calculated based on the method described previously [32]. All isolates were measured in triplicates to obtain the mean representing each isolate. The statistical comparison between studied groups was made using the mean from five isolates from each group.

### 2.3. Apoptotic Characterization

A concentration of 1 × 10^5^ cells/mL parasites from Day 3 cultures were inoculated into a 1 mL medium and incubated for 96 h. The cell viability and percentage of apoptotic forms were determined every 24 h. Cells were harvested, washed twice with phosphate-buffered saline, and stained using the Apoptotic, Necrotic and Healthy Cell Quantification Kit (Biotium Inc., Hayward, CA, USA), following the manufacturer’s protocol. Fluorescein isothiocyanate (FITC)-Annexin V provided in the kit stains the apoptotic cell green by binding to phosphatidylserine translocated to the cell surface membrane. Stained cells were observed under the Olympus BX 51 epifluorescence microscope (Olympus, Hamburg, Germany) using image analyzer software. The percentage of apoptotic cells was determined by randomly counting the number of apoptotic cells per 100 cells.

### 2.4. Ultrastructure Analysis

Parasites from Day 3 cultures were pooled and washed three times with PBS pH 7.4 and centrifuged at 1000 rpm for 5 min. The pelleted cells were re-suspended in 4% glutaraldehyde in 0.1 M sodium cacodylate buffer, pH 7.3 at 4 °C, washed thoroughly with cacodylate buffer, and post-fixed for 30 min in 1% osmium tetroxide in cacodylate buffer. The fixed cells were dehydrated in ascending series of ethanol and embedded in epoxy resin. Semi-thin sections were stained with toluidine blue. Ultrathin sections were cut using an ultramicrotome, contrasted with uranyl acetate and lead citrate, and viewed using a transmission electron microscope (LEO Libra 120) [32].

### 2.5. Drug Resistance

The stock solution of metronidazole (MTZ) was prepared at 10 mg/mL as described in the previous study [33]. Cells were harvested from the Day 3 culture, washed with phosphate buffer saline (PBS) twice, and inoculated into a fresh medium. The final volume was adjusted to 1 mL by adding MTZ to achieve a final concentration of 1 mg/mL and 0.1 mg/mL. The cells were incubated for ten days at 37 °C and counted every 24 h using the trypan blue exclusion test. The experiment was done in triplicates for each isolate.

### 2.6. Viability in Distilled Water

Parasites from Day 3 cultures were harvested and washed thoroughly three times and counted to prepare an inoculum, with a final concentration of 1 × 10^6^ cells/mL. The cells were inoculated into 1ml of sterile distilled water. The cells were observed every 15 min for 3 h. The number of viable cells was determined using the trypan blue exclusion test.

### 2.7. Purification of Blastocystis sp. and Extraction of Solubilized Antigens

*Blastocystis* sp. cells were purified from bacteria-contaminated culture using density gradient centrifugation. The cells were pooled into one culture tube and washed twice with phosphate-buffered saline (PBS) for 5 min at 1000 rpm. Five milliliters of the cell suspension were then layered carefully onto 6 ml of Ficoll–Paque without agitation. It was then spun for 20 min at 1800 rpm. *Blastocystis* sp. Cells with minimal bacterial contaminants found right above the thick layer of yellowish-white clump was then gently isolated and washed with PBS. The antigens were extracted using the freeze-thaw method and stored at −20 °C until further use.

### 2.8. Colorimetric Assay for the Quantification of Protease Activity

Protease activity was quantified through a colorimetric assay with azocasein as a substrate. A solution of 1.5 mg/mL azocasein was prepared with PBS and stored at 4 °C for less than 2 weeks. The final concentration of cell lysates was fixed at 0.1 mg/mL. Azocasein assay was carried out, the optical density reading was done at 442 nm, and the reading was calculated based on a previous study [20].

### 2.9. Inhibition Assay to Determine Specific Protease Activity

Proteases typically exist in a mixture of cysteine, serine, aspartic, and metalloprotease types. Different protease inhibitors were used to detect the dominant type of protease present. Protease inhibitors used were E-64 (cysteine protease inhibitor), phenylmethanesulfonyl fluoride (PMSF) (serine protease inhibitor), pepstatin A (aspartic protease inhibitor), and EDTA (metalloprotease inhibitor). All the inhibitors were obtained from Sigma-Aldrich. Protocol used in quantification of protease was applied with supplementation of protease inhibitors at a particular concentration as reported by previous studies [21,34].

### 2.10. Colon Cell Culture and Proliferation Studies

Human colon cancer cells (HCT 116) were grown in RPMI medium supplemented with L-Glutamine, antibiotics, and fetal bovine serum (FBS). Cells were transferred to a 96-well plate with a concentration of 1000 cells in 100 μL per well. Solubilized antigens of *Blastocystis* sp. from urban and orang asli individuals were introduced at a final concentration of 0.01 μg/mL. Cell proliferation was analyzed through MTT assay as described previously [26].

### 2.11. Statistical Analysis

All the experiments for growth characterization and colon cell proliferation studies involved 20 isolates (5 symptomatic and 5 asymptomatic from urban and orang asli, respectively). The mean from three technical replicates represented each isolate. A deviation from the standardized limit was considered an anomaly for the technical replicates. Statistical analysis was done using students’ *t*-tests to compare the mean (from five isolates of each group) using SPSS (IBM SPSS Statistics for Macintosh, Version 21.0. Armonk, NY, USA). A *p*-value of less than 0.05 was regarded as significant in this study.

## 3. Results

### 3.1. Blastocystis sp. ST3 Phylogenetic Analysis and Growth Profile

In this study, only *Blastocystis* sp. ST3 from symptomatic and asymptomatic isolates of urban and orang asli individuals was selected. Phylogenetic analysis of the 18S rDNA sequences showed close similarity (97–99%) among the ST3 symptomatic and asymptomatic isolates obtained from urban and orang asli individuals (CLUSTALW). The phylogenetic tree developed using a neighbor-joining method demonstrated no specific cluster formation within the ST3 sequence of urban and orang asli isolates (Appendix A). Growth profile of the *Blastocystis* sp. ST3 cells in vitro showed a greater cell number in symptomatic and asymptomatic orang asli isolates. The peak cell counts of *Blastocystis* sp. from urban individuals were 2.43 × 10^6^ cells/mL in symptomatic isolates and 3.97 × 10^6^ cells/mL in asymptomatic isolates. In parasites obtained from orang asli individuals, the peak cell count was two-fold higher in both symptomatic and asymptomatic isolates at 3.8 × 10^6^ cells/mL and 5.38 × 10^6^ cells/mL, respectively. Parasites isolated from orang asli showed lower generation time in both symptomatic and asymptomatic isolates. A significant difference in generation time was seen only between orang asli symptomatic and asymptomatic isolates (t_(8)_ = 2.49; *p* = 0.019) (Figure 1B,D).

### 3.2. Growth and Morphometric Assessment

The difference in the number of granular forms was seen only between symptomatic isolates of urban and orang asli individuals. *Blastocystis* sp. from symptomatic orang asli isolates possessed a higher number of granular forms compared to asymptomatic ones (t_(8)_ = 2.18, *p* = 0.03). There was an obvious correlation between the number of granular forms (Figure 2A,B) and apoptotic forms (Figure 3) seen in *Blastocystis* sp. isolates of urban and orang asli. In general, isolates obtained from the orang asli population (both symptomatic and asymptomatic) with a greater number of granular forms showed significantly higher apoptotic forms when compared to urban isolates (t_(8)_ = 1.97, *p* = 0.042; t_(8)_ = 2.13, *p* = 0.034). On the other hand, amoebic forms predominated urban symptomatic isolates of *Blastocystis* sp. with up to 1.7 × 10^5^ cells/mL of amoebic forms. Very low numbers of amoebic forms (1.6 × 10^4^ cells/mL) were seen in orang asli *Blastocystis* sp. Isolates, and a complete absence of amoebic forms was noticed in urban asymptomatic isolates (Figure 2C,D).

### 3.3. Ultrastructural Assessment

The transmission electron microscopy (TEM) images showed that parasites isolated from urban individuals predominantly possessed a surface fuzzy coat that was not seen in parasites from orang asli settlements. The fuzzy coat was found intact in urban symptomatic isolates (Figure 4A–C) but was detached in asymptomatic ones (Figure 4D–F). There was prominent electron-dense material in *Blastocystis* sp. from urban symptomatic and asymptomatic isolates whereas orang asli isolates demonstrated well-rounded cells with a large central vacuole surrounded by cytoplasm and prominent nucleus (Figure 4G–J).

### 3.4. Assessment of Resistance to Metronidazole and Viability in Distilled Water

Orang asli symptomatic isolates showed greater viability at 0.1 mg/mL and 1.0 mg/mL of metronidazole compared to *Blastocystis* sp. from urban isolates (Figure 5A–D). However, in general, symptomatic isolates show higher viability than asymptomatic isolates in *Blastocystis* sp. obtained from urban and orang asli individuals. Isolates obtained from orang asli were again shown to be robust and resisted lysis in distilled water compared to urban isolates (Figure 5E,F).

### 3.5. Quantification of Protease Activity Using Colorimetric and Inhibition Assay

In general, all orang asli isolates (symptomatic and asymptomatic) showed significantly greater protease activity, respectively, compared to isolates obtained from urban individuals (t_(8)_ = 2.15, *p* = 0.032; t_(8)_ = 2.10, *p* = 0.034). Among orang asli isolates of *Blastocystis* sp., the mean protease activity was insignificant between symptomatic and asymptomatic isolates. Significantly higher protease activity was seen in urban symptomatic isolates than in asymptomatic ones (t_(8)_ = 1.93, *p* = 0.045). Protease activity assayed with an array of protease inhibitors indicated the predominance of specific protease. Urban symptomatic isolates demonstrated a significant predominance of cysteine protease activity (inhibition to E64) (t_(8)_ = 2.04, *p* = 0.038). There was no significant inhibition found with other protease inhibitors. *Blastocystis* sp. protease obtained from asymptomatic urban individuals showed a significant inhibition towards PMSF (t_(8)_ = 2.34, *p* = 0.023) and pepstatin A (t_(8)_ = 2.16, *p* = 0.031), indicating the presence of serine and aspartic protease. Symptomatic orang asli individuals only showed a significant predominance of cysteine protease (t_(8)_ = 2.78, *p* = 0.012). Other inhibitors failed to inhibit protease activity significantly. Proteases isolated from asymptomatic orang asli persons demonstrated greater inhibition towards E64 (t_(8)_ = 1.952, *p* = 0.043) and PMSF (t_(8)_ = 1.97, *p* = 0.042), implying the predominance of cysteine and serine protease (Figure 6).

### 3.6. Blastocystis sp.-Induced Proliferation of Cancer Cells

Cell proliferation was seen significantly higher when the cancer cells were treated with urban symptomatic isolates of *Blastocystis* sp. compared to urban asymptomatic isolates (t_(8)_ = 2.36, *p* = 0.023) and orang asli isolates (t_(8)_ = 2.42, *p* = 0.021; t_(8)_ = 2.51, *p* = 0.015) (Figure 7). There was approximately three times greater stimulation of proliferation when cancer cells were treated with antigens belonging to *Blastocystis* sp. isolated from urban symptomatic individuals, which were 60%. The proliferation rate was significantly low when cancer cells were treated with solubilized antigens of orang asli isolates of *Blastocystis* sp. No significant difference in proliferation was seen between the treatment of antigens from *Blastocystis* sp. isolated from symptomatic and asymptomatic orang asli individuals.

## 4. Discussion

*Blastocystis* sp. is an intestinal protozoan parasite that lives by closely interacting with its microbial environment [35]. Studies have reported that *Blastocystis* sp. is a common and stable component of the human intestinal microbiome [18]. Microbiome reportedly varies depending on the host factors such as diet, health issues, and lifestyle in a population [36]. However, there is a scarcity of information on variation in *Blastocystis* sp. when isolated from a population with a distinct gut microbiome.

Previously, *Blastocystis* sp. isolated from patients with gastrointestinal illnesses showed differences not only in protein patterns [37,38] but also in molecular characterization using PCR amplification and restriction analysis [39]. In studies involving a single subtype, ST3, the phenotypic difference between symptomatic and asymptomatic isolates of *Blastocystis* sp. was observed [40]. Another study on ST3 demonstrated phenotypic variability in *Blastocystis* sp. isolated from IBS and non-IBS individuals. This was suggested to be due to the alteration in the microbiome [32]. However, these studies were only restricted to parasites isolated from the urban population, while a high prevalence of *Blastocystis* sp. is usually seen in the rural population [10]. In recent years, rapid urbanization has driven the migration of rural individuals and ultimately increased the risk of transmission to the urban population. Hence, recognizing the phenotypic diversity of *Blastocystis* sp. from different populations enables the identification and prevention of potentially harmful strains and their spread [41]. In the present study, *Blastocystis* sp. ST 3 isolated from urban and orang asli populations was studied, and significant phenotypic differences were demonstrated.

Studies carried out previously on apoptosis in *Blastocystis* sp. have suggested that when cells become apoptotic, a mechanism is triggered to release larger numbers of granular forms as a mode of survival [42]. The current study demonstrated that *Blastocystis* sp. obtained from orang asli individuals (symptomatic and asymptomatic) had higher cell numbers and concurred with the previous study where there was a higher apoptosis rate. Higher parasite cell numbers could have arisen from the granular form that produces viable progeny. Previously, Ragavan et al., 2014 showed the presence of a thicker surface coat in *Blastocystis* sp. isolated from IBS patients, which has been postulated previously to facilitate adherence of bacteria to the surface coat [43]. Amoebic forms of *Blastocystis* sp. have been reported to present mainly in symptomatic *Blastocystis* sp. isolates [44]. In the present study, larger numbers of amoebic forms with attached fuzzy surface coats observed in urban symptomatic isolates of *Blastocystis* sp. suggest an enhanced adherence of bacteria, possibly resulting in elevated microbiome targeted-pathogenic potential as *Blastocystis* sp. is known to prey on bacteria [45]. It is possible that the prolonged presence of the *Blastocystis* sp. interacting with a particular host microbial environment in asymptomatic urban individuals and orang asli individuals could have caused detachment and loss of the fuzzy coat. In these instances, *Blastocystis* sp. could behave as a commensal of a healthy gut compared to an opportunistic parasite in most symptomatic infections.

Symptomatic *Blastocystis sp*. ST3 has been reported to show greater resistance towards metronidazole when compared with asymptomatic isolates [46]. However, the study was silent on where the source of *Blastocystis* sp. was isolated from. The present study demonstrates that orang asli isolates of *Blastocystis* sp. showed resistance even at a high concentration of metronidazole (1 mg/mL). In this study, *Blastocystis* sp. was isolated from individuals who have never consumed any antibiotics or antiprotozoal medication. This eliminates the possibility of resistance gained prior to isolation and corroborates that the difference seen is acquired from the environment. Another study reported the ability of *Blastocystis* sp. ST 3 to withstand the effect of rural river water during wet and dry season [47]. Generally, vacuolar forms of *Blastocystis* sp. lyses in distilled water (unpublished observation). The cyst has been shown to withstand up to 19 days in water [48] but this study is the first to show that orang asli *Blastocystis* sp. possess a greater ability to resist lysis in water, suggesting the possible presence of a robust wall even in vacuolar forms. It is unknown if these forms are responsible for prolonged colonization. More ultrastructural studies on cells isolated from different populations should explain the variation seen.

Urban and orang asli populations reportedly have wide differences in terms of the gut microbial population [15]. Reports have suggested that protozoan parasites such as Entamoeba histolytica and Trichomonas vaginalis may well adapt to the environment and the expression of related diseases could be dependent on the composition of its microbial environment [49,50]. Similarly, *Blastocystis* sp. may experience phenotypic alterations due to the influence of different gut environments in urban and orang asli. To date, no studies have highlighted this variation in *Blastocystis* sp. and its implications. Previously, a study has suggested that ascribing pathogenic status to a specific subtype may be an over-generalized attempt due to the vulnerability of *Blastocystis* sp. to gut environment in normal and diseased conditions [32]. The present study concurs as there are significant differences seen when *Blastocystis* sp. ST3 is isolated from urban and orang asli populations, which could be likely due to the difference in gut environment as well. This indicates that there is a strong influence of environment on the phenotype of *Blastocystis* sp.

Proteases have been reported to play important role in the infection of parasites such as *Entamoeba histolytica*, *Plasmodium* sp., *Acanthamoeba* sp., and *Trichomonas vaginalis* [34,51,52]. Proteases from *Blastocystis* sp. have been reported to play similar pathogenic roles such as causing intestinal barrier disruption, degradation of immunoglobin A (IgA), and elicitation of inflammatory cytokines [22,23]. Previous studies have even reported protease activity from different subtypes, namely ST 7 and ST 4 [53]. However, the present study focused only on ST 3, as it is more clinically prevalent and frequently seen colonizing the human gut in the Malaysian population [10].

*Blastocystis* sp. has been reported in the past to harbor predominant protease belonging to the cysteine class; later studies proved that these proteases were the cathepsin and legumain type [21,54]. This study concurred with the previous findings that the class of protease seen was cysteine. The predominance of serine protease and aspartic protease was also seen in asymptomatic isolates in general. The findings suggest the association of elevated cysteine protease to symptomatic infection and low levels of cysteine protease could give rise to asymptomatic infection. Serine proteases have been implicated to be responsible for asexual reproduction in apicomplexans and encystation in protozoans [55,56]. Similarly, in asymptomatic isolates, the presence of serine protease could have facilitated the production of high cell numbers seen.

Studies have reported an increased rate of colon cancer cell proliferation exclusively when treated with solubilized antigens of *Blastocystis* sp. isolated from symptomatic individuals and ST 3 [25,26]. This observation suggests that the proliferation of cancer cells in vitro could be an indication to differentiate pathogenic properties in *Blastocystis* sp. The current study analyzed the effect of solubilized antigen extracted from *Blastocystis* sp. ST 3 when isolated from urban and orang asli individuals. We observed that only isolates obtained from symptomatic urban individuals showed significant proliferation. A report on the microbiome of the urban population demonstrated a greater abundance of pathogenic bacteria and cellular processes that are closely associated with inflammation and colorectal cancer development [57]. A greater number of amoebic forms seen in *Blastocystis* sp. from symptomatic urban individuals could attract and engulf more of these bacteria, which could have possibly resulted in changes to antigenic content of solubilized antigens compared to isolates obtained from asymptomatic urban and orang asli individuals. However, this postulation of whether surrounding bacteria affect the antigenic composition of *Blastocystis* sp. needs more studies to be validated.

## 5. Conclusions

In the present study, the finding implicates that *Blastocystis* sp. may undergo alteration while adapting to its microenvironment and potential pathogenic property in this parasitic organism may be influenced by the surrounding bacteria. Our study is limited to only the assessment of *Blastocystis* sp. cells and does not include the study on accompanying bacteria and its influence when the composition is altered. However, the distinct phenotypic difference in a single subtype seen in this study reveals that the gut microenvironment does affect the phenotype, which potentially contributes to the confusion in the pathogenicity of *Blastocystis* sp. (Figure 8). Our study also confirms that the parasite isolation source can play a very important role. Therefore, general assumptions about the potential pathogenicity of *Blastocystis* sp. genotypes should be avoided, as it can be dramatically modulated by the parasite isolation source.

## Figures and Tables

**Figure 1 biology-11-01211-f001:**
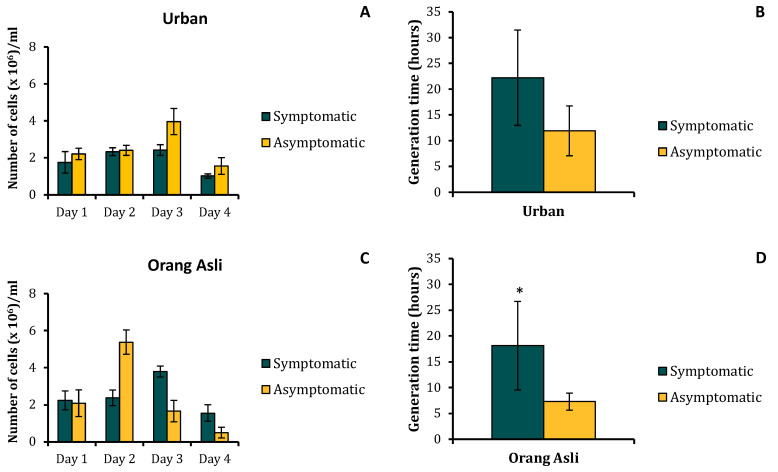
Growth profile of *Blastocystis* sp. isolated from urban and orang asli individuals (**A**,**C**) with their respective generation time (**B**,**D**). Viable cells were counted in 1 mL volume using the trypan blue exclusion test. Values are expressed as mean ± SD. * *p* < 0.05 for Students *t*-test.

**Figure 2 biology-11-01211-f002:**
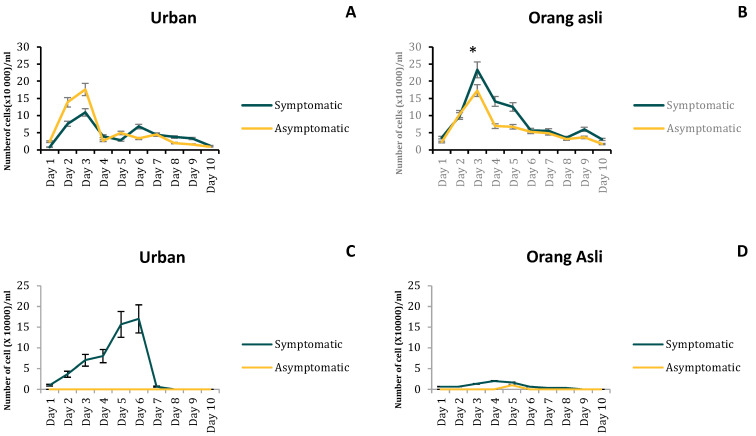
The average number of granular forms (**A**,**B**) and amoebic forms (**C**,**D**) of *Blastocystis* sp. isolated from urban and orang asli individuals. Values are expressed as mean ± SD. * *p* < 0.05 is the comparison between granular forms of symptomatic isolates.

**Figure 3 biology-11-01211-f003:**
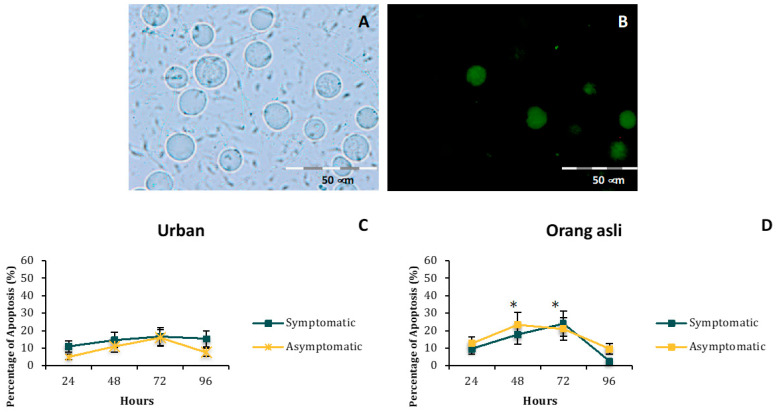
The FITC Annexin V stained apoptotic forms exhibited by isolates obtained from urban and orang asli individuals (**A**,**B**) and their percentages seen for 96 h (**C**,**D**). Green-stained cells in plot B indicate the apoptotic forms at 24 h of incubation when stained using FITC Annexin V. Values are expressed as mean ± SD. * *p* < 0.05 for comparison between peak percentage apoptotic forms between parasites isolated from urban and orang asli individuals.

**Figure 4 biology-11-01211-f004:**
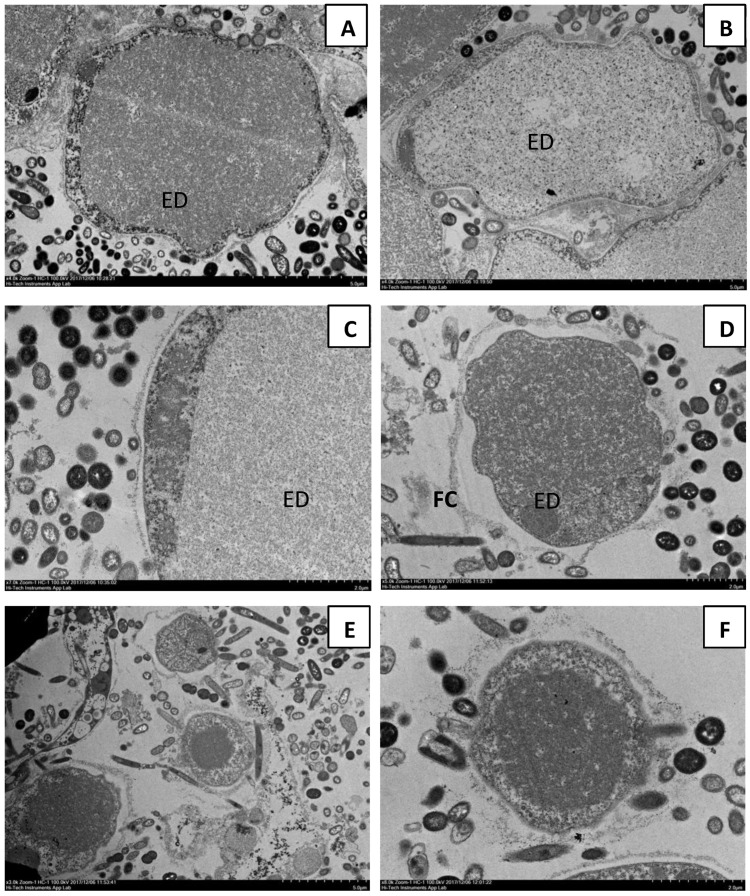
Transmission electron microscopy (TEM) comparison between *Blastocystis* sp. isolated from urban and orang asli individuals. (**A**–**C**) Isolates obtained from symptomatic urban individuals with an intact fuzzy coat (FC). (**D**–**F**) Parasite isolated from asymptomatic urban individuals showing detached FC. (**G**,**H**) Isolates obtained from symptomatic orang asli individuals with rounded cells and prominent central vacuole. (**I**,**J**) Isolates obtained from asymptomatic orang asli individuals with a prominent central vacuole. Note: (ED) Electron-dense material.

**Figure 5 biology-11-01211-f005:**
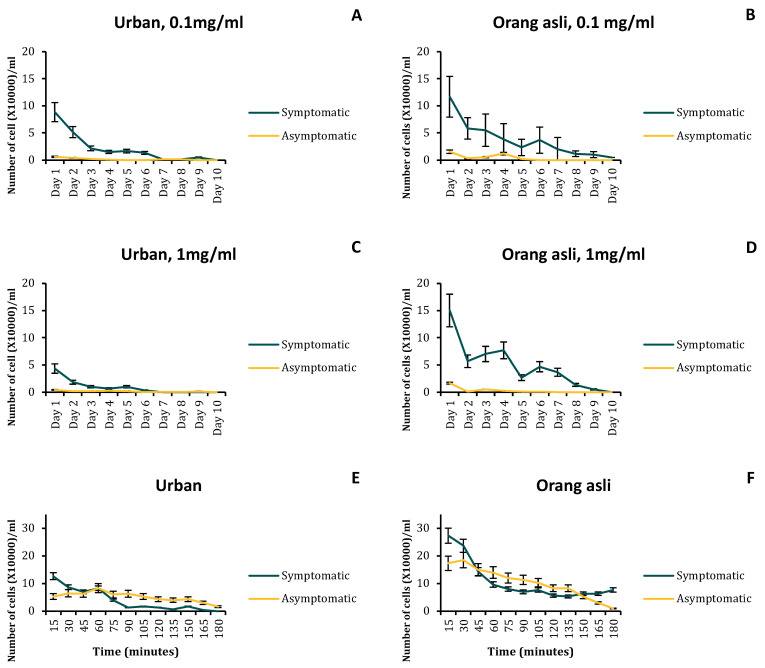
Resistance towards metronidazole (MTZ) and distilled water in vitro. (**A**–**D**) The growth profile of *Blastocystis* sp. was treated with MTZ at 0.1 mg/mL and 1 mg/mL. (**E**,**F**) *Blastocystis* sp. obtained from urban and orang asli inoculated in distilled water. *Blastocystis* sp. isolated from orang asli individuals showed greater robustness than parasites obtained from urban individuals. Values are expressed as mean ± SD.

**Figure 6 biology-11-01211-f006:**
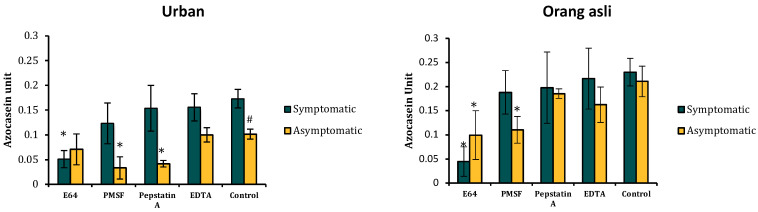
The specific type of protease activity of *Blastocystis* sp. isolated from urban and orang asli individuals. Note the predominance of cysteine protease in symptomatic and serine protease in asymptomatic isolates. Values are expressed as mean ± SD. * *p* < 0.05, Student’s *t*-test comparing the mean to control. # *p* < 0.05, Student’s *t*-test comparing the mean protease activity between symptomatic and asymptomatic.

**Figure 7 biology-11-01211-f007:**
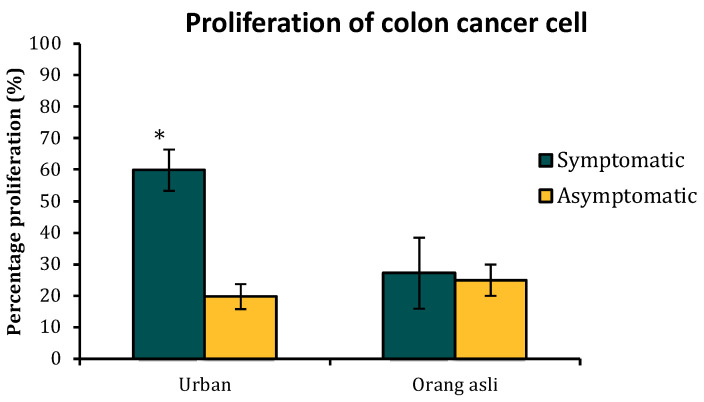
The proliferation of HCT 116 colon cancer cells when treated with solubilized antigens from *Blastocystis* sp. isolated from urban and orang asli individuals. Colon cells at the density of 1000 cells in 100 μL were treated with 0.01 μg/mL of antigen. The proliferation of cancer cells was seen only with isolates obtained from urban symptomatic individuals. Values are expressed as mean ± SD. * *p* < 0.05 in *t*-test for comparison within urban isolates and between urban and orang asli isolates.

**Figure 8 biology-11-01211-f008:**
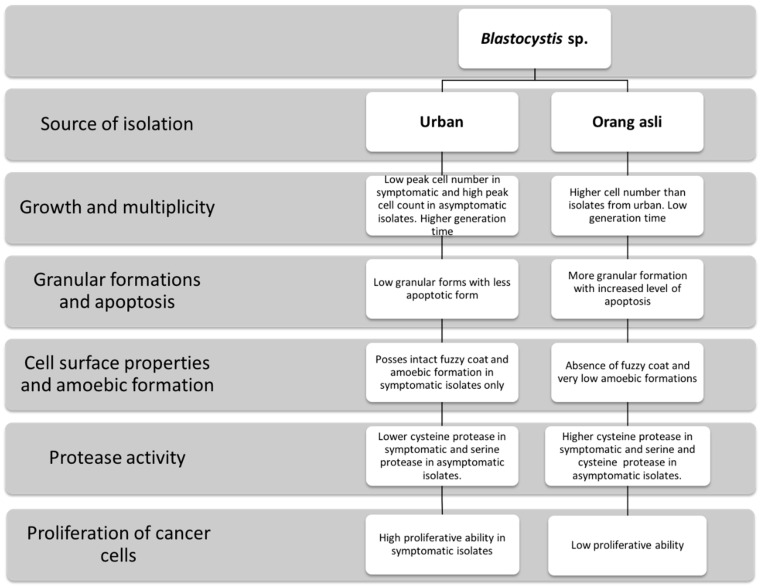
Schematic diagram explaining the phenotypic and potential pathogenic differences between *Blastocystis* sp. isolated from urban and orang asli individuals. These groups have distinct gut microbial profile. It is clear that gut environment plays a pivotal role in altering the phenotype of *Blastocystis* sp. and its pathogenic potentials.

## Data Availability

The data generated and analyzed during this study are not publicly available as it would violate the agreement with the funding agency, and the data used here are a part of a larger study that is still ongoing. Hence, the confidentiality of the data at this point in time is important to us. However, the data are available on reasonable request for non-commercial purposes from the corresponding author.

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
