# Peer review of "Distinct Phenotypic Variation of Blastocystis sp. ST3 from Urban and Orang Asli Population—An Influential Consideration during Sample Collection in Surveys"

_biology, 2022, doi:10.3390/biology11081211_

Round 1
Reviewer 1 Report
The reviewer is convinced that the authors are providing reasonable and sound information about an interesting and important protozoon. There must, however, a lot been done prior to recommending the manuscript for publication.
* Grammar must be checked throughout. Only examples from the 'Abstract': "Phenotypic variation ... suggest ..." must be 'suggests; "... "the phenotypic features .... is unknown." must be 'are unknown'. "Pathogenic potentials ..." must read 'Pathogenic potential' and needs singular form of the verb. There is a lot more throughout the text.
* Please check carefully the use of articles. It could be helpful to have the text read by someone growing up with Englisch or American as mother tongue.
* Avoid unnecessary abbreviations throughout. Keep in mind that publications are often read by colleagues that are not specialists in the field. The authors should make reading as easy as possible. Please avoid especially abbreviations that are used only a few times. In any case explain abbreviations when first used in a given paragraph. There are several instances in the text. This one is an example: "... with and without GI symptoms.", another one is "Hg-cultures". Please be more precise and carefully avoid lab slang expressions.
* Please remember that there must always be a space between numbers and their dimension.
* Check sources and provenience of material carefully. An example: Olympus miscroscopes are not made in Wetzlar. Wetzlar is associated with Leitz. Olympus is a Japanese company with buros throughout the world. The German one is in Hamburg. All this, and other points, may seem of little importance, but what should readers think about solidity and soundness of a publication where already minor points - inncluding language - have been treated without appropriate diligence?
* Very important: The authors must always make very clear, what they are talking about. An example: "The sequences showed high similarity, around 97-99 %" cannot be understood. What sequences are the authors talking about? What has been sequenced, what has been compared? Would it not be helpful to add a little table that makes similarities clear?
* Please use reasonable numbers at axes. "8000000" and similar is simply not readable at a first glance. Very important: Figure captions must contain everything that lets the reader understand the figure without constant reference to the main text.
* Could the authors please provide pictures of typical results from apoptosis determinations? What do the percantages in Figure 3 really mean? There must be more detailed and maybe critical information. This would be very helpful.
* The text must be restructured, figures, figure captions etc, sometimes are shown in a very unusual and unexpected order.
Overall the reviewer thinks that the authors provide a sound scientific approach with probably important results and reasonable discussion and interpretation. However, prior to publication, a very thorough revision including re-writing of several parts is required.
The points above, there could be even more, are meant to be helpful towards improving the manuscript. The reviewer will be glad to recommend the manuscript for publication after very careful revision.
Author Response
We appreciate the reviewer's comments. Please see the attachment for the response.

Reviewer 2 Report
The submitted manuscript by Rajamanikam et al. is an interesting approach to dilucidate the phenotypic variation of Blastocystis sp. ST3 and to reach a better understanding of its pathogenicity. Although this study presents some limitations, such as the inclusion of data from the concomitant bacteria and its influence, I consider that this paper deserves consideration to be published.
The major concerns I could find are mainly format issues thorough the article:
Line 78: Indicate that GI refers to gastrointestinal here, instead of line 80.
Line 95: Reference 53 without previous mention to references 27-52. Therefore, the list of references has to be remade from this point. Reference 53 should be reference 27 and so on.
Lines 187-198 and 240-244: Different format of text
Line 211: Figure 3 is cited before Figure 2
Figure 1: A, B, C and D can’t be read properly in the manuscript
Figure 5: There is a square image that shouldn’t be there
Line 259-262: Can’t be read properly
Author Response

(The authors gave the same response as above.)

Reviewer 3 Report
The manuscript by Arutchelvan Rajamanikam et al. must be further improved before resubmitting for consideration. The figures are not well-organized and some of the panels are not correctedly labelled. The English of the manuscript must be further improved. Some figures and the main text are overlapped. The authors are too careless. The authors must revise very very carefully.
Author Response

(The authors gave the same response as above.)

Round 2
Reviewer 1 Report
The reviewer has no substantial points of criticism any more.
Reviewer 3 Report
The authors have revised the manuscript. It can be considered for publication.